# Rice Bran Reduces Weight Gain and Modulates Lipid Metabolism in Rats with High-Energy-Diet-Induced Obesity

**DOI:** 10.3390/nu11092033

**Published:** 2019-08-30

**Authors:** Suh-Ching Yang, Wen-Ching Huang, Xin Er Ng, Mon-Chien Lee, Yi-Ju Hsu, Chi-Chang Huang, Hai-Hsin Wu, Chiu-Li Yeh, Hitoshi Shirakawa, Slamet Budijanto, Te-Hsuan Tung, Yu-Tang Tung

**Affiliations:** 1School of Nutrition and Health Sciences, Taipei Medical University, Taipei 110, Taiwan; 2Department of Exercise and Health Science, National Taipei University of Nursing and Health Sciences, Taipei City 112, Taiwan; 3Graduate Institute of Metabolism and Obesity Sciences, Taipei Medical University, Taipei 110, Taiwan; 4Graduate Institute of Sports Science, National Taiwan Sport University, Taoyuan 333, Taiwan; 5Graduate School of Agricultural Science, Tohoku University, Miyagi 980-8572, Japan; 6Faculty of Agricultural Engineering and Technology, Bogor Agricultural University, Jawa Barat 16680, Indonesia; 7Nutrition Research Center, Taipei Medical University Hospital, Taipei 110, Taiwan; 8Cell Physiology and Molecular Image Research Center, Wan Fang Hospital, Taipei Medical University, Taipei City 116, Taiwan

**Keywords:** high-energy diet (HED), lipid metabolism, lipidomics, obesity, rice bran

## Abstract

Obesity has become an epidemic worldwide. It is a complex metabolic disorder associated with many serious complications and high morbidity. Rice bran is a nutrient-dense by product of the rice milling process. Asia has the world’s highest rice production (90% of the world’s rice production); therefore, rice bran is inexpensive in Asian countries. Moreover, the high nutritional value of the rice bran suggests its potential as a food supplement promoting health improvements, such as enhancing brain function, lowering blood pressure, and regulating pancreatic secretion. The present study evaluated the anti-obesity effect of rice bran in rats with high-energy diet (HED)-induced obesity. Male Sprague–Dawley rats were randomly divided into one of five diet groups (*n* = 10 per group) and fed the following for eight weeks: Normal diet with vehicle treatment, HED with vehicle, rice bran-0.5X (RB-0.5X) (2% wt/wt rice bran), RB-1.0X (4% wt/wt rice bran), and RB-2.0X (8% wt/wt rice bran). Rice bran (RB-1.0X and RB-2.0X groups) markedly reduced obesity, including body weight and adipocyte size. In addition, treating rats with HED-induced obesity using rice bran significantly reduced the serum uric acid and glucose as well as the liver triglyceride (TG) and total cholesterol (TC). Furthermore, administration of an HED to obese rats significantly affected hepatic lipid homeostasis by increasing phosphotidylcholine (PC; 18:2/22:6), diacylglycerol (DG; 18:2/16:0), DG (18:2/18:1), DG (18:1/16:0), cholesteryl ester (CE; 20:5), CE (28:2), TG (18:0/16:0/18:3), and glycerol-1-2-hexadecanoate 3-octadecanoate. However, the rice bran treatment demonstrated an anti-adiposity effect by partially reducing the HED-induced DG (18:2/18:1) and TG (18:0/16:0/18:3) increases in obese rats. In conclusion, rice bran could act as an anti-obesity supplement in rats, as demonstrated by partially reducing the HED-induced DG and TG increases in obese rats, and thus limit the metabolic diseases associated with obesity and the accumulation of body fat and hepatic lipids in rats.

## 1. Introduction

The prevalence of obesity is rising worldwide. If this trend continues, 57.8% of the adult population worldwide (3.3 billion people) may be overweight or obese by 2030 [1]. Obesity is a serious global health problem leading to many health complications, such as type 2 diabetes, fatty liver disease, dyslipidemia, hypertension, coronary artery disease, stroke, heart failure, reproductive and gastrointestinal cancers, osteoarthritis, obstructive sleep apnea, and gallstones [2,3]. However, most of the approved and marketed anti-obesity drugs have been withdrawn due to serious side effects [4,5,6]. Therefore, natural products such as anti-obesity agents are indispensable for combating obesity.

Rice is the second leading cereal crop, and the highest rice production is found in Asia (90% of the world’s total rice production) [7]. Therefore, in Asian countries, rice bran (RB) is inexpensive. RB, one of the most abundant byproducts produced in the rice milling industry, contains appreciable quantities of nutrients (protein, fat, unsaturated fatty acids, dietary fiber, K, Ca, Mg, and Fe) and antioxidants (γ-oryzanol, tocopherols, tocotrienols, and ferulic acid) [7,8,9,10,11]. Due to its nutritional value, it can also be used as a health food supplement that improves brain function, lowers blood pressure and cholesterol concentration, and regulates pancreatic secretion [12,13].

The lipidomic analysis provides a powerful approach for assessing a wide range of lipid species in biological systems. The liquid chromatography-mass spectrometry (LC-MS) was successfully applied to determine the metabolomic alternations in the plasma or serum of rodents fed with a high-fat and high-carbohydrate (fructose and sucrose) diet [14]. We investigated the anti-obesity effect of RB through an ultraperformance liquid chromatography coupled with mass spectrometry (UPLC-QTOF/MS) in diet-induced-obesity rat models by determining the serum lipid profiles (i.e., lipidomics).

## 2. Materials and Methods

### 2.1. Animals and Study Design

Fifty male Sprague–Dawley (SD) rats (six weeks old; 200–250 g) obtained from BioLASCO (A Charles River Licensee Corp., Yi-Lan, Taiwan) were used in this study. The rats were maintained in a laboratory under controlled conditions of a 12-h light/dark cycle, 65% ± 5% relative humidity, and 24 °C ± 2 °C. The care and use of animals followed the guidelines of the Institutional Animal Care and Utilization Committee of the National Taiwan Sport University (IACUC-10707). The animals were acclimatized to the laboratory conditions for two weeks.

After two weeks of acclimatization, all 50 rats were randomly divided into one of five treatment groups based on diet: (1) Chow diet (control; *n* = 10), (2) High-energy diet (HED/control; *n* = 10), (3) HED and RB-0.5X (HED/0.5X; *n* = 10), (4) HED and RB-1.0X (HED/1.0X; *n* = 10), or (5) HED and RB-2.0X (HED/2.0X; *n* = 10). The body weight change and diet consumption were measured every week. The rats were sacrificed after eight weeks of intervention. The compositions of each diet were as follows: The chow diet (LabDiet Rodent 5001) contained 3.35 kcal/g with 28.5% protein, 13.4% fat, and 58.1% carbohydrates. The HED contained 4.22 kcal/g with 8% (wt/wt) lard oil, 44% (wt/wt) high-fructose syrup, and 48% (wt/wt) standard chow. The RB-0.5X diet contained 8% (wt/wt) lard oil, 44% (wt/wt) high-fructose syrup, 46% (wt/wt) standard chow, and 2% (wt/wt) RB for 4.24 kcal/g. The RB-1.0X diet contained 8% (wt/wt) lard oil, 44% (wt/wt) high-fructose syrup, 44% (wt/wt) standard chow, and 4% (wt/wt) RB for 4.26 kcal/g. The RB-2.0X diet contained 8% (wt/wt) lard oil, 44% (wt/wt) high-fructose syrup, 40% (wt/wt) standard chow, and 8% (wt/wt) RB for 4.29 kcal/g. At the end of the experiment, all rats were fasted for 12 h and anesthetized, and blood samples were collected through a cardiac puncture. The serum was obtained by centrifugation at 1500× *g* 4 °C for 10 min. The liver, omental fat, and epididymal fat were resected and immediately weighed. Additionally, the pathological histology of epididymal fat and liver tissues was performed. All of the samples were snap-frozen and stored at −80 °C until further analysis.

### 2.2. Determination of Biochemical Markers

The collected blood samples were used to detect the serum levels of the uric acid, glutamate oxaloacetate transaminase (GOT), free fatty acid (FFA), total cholesterol (TC), triglyceride (TG), high-density lipoprotein (HDL), low-density lipoprotein (LDL), and glucose content using an autoanalyzer (Hitachi 7060, Company, City, Country). Total lipids were extracted from the liver through the Folch method [15]. For the liver TC and TG determination, 20 mg of liver tissue was homogenized in a 200 μL solvent (chloroform:isopropanol:nonyl phenoxypolyethoxylethanol, NP40 = 7:11:0.1). Centrifuged at 12,000× *g* for 10 min, an aliquot of 100 μL was extracted and dried. The pellet was reconstituted with a buffer (1 M of potassium phosphate, pH = 7.4, 500 mM of sodium chloride, and 50 mM of cholic acid), and a water bath sonication was employed to dissolve the precipitate [16]. The cholesterol fluorometric assay kit (Cayman, Ann Arbor, MI, USA) and colorimetric triglyceride assay kit (Cayman, Ann Arbor, MI, USA) were further applied to the liver TC and TG contents analysis. 

### 2.3. Hematoxylin and Eosin Staining

The liver tissue and epididymal fat were fixed in a 10% buffered formaldehyde. The tissues were then soaked in absolute ethanol overnight, embedded in paraffin, and cut into 4-μm-thick slices. Sections were then stained with hematoxylin and eosin and examined by a clinical pathologist under a light microscope equipped with a CCD camera (BX-51, Olympus, Tokyo).

### 2.4. Serum Lipidomics Analyses

#### 2.4.1. Serum Lipid Extraction

The Folch method [13] was used with a slight modification for extraction of the serum lipids from six samples of the vehicle, HED, and 2.0X groups. Briefly, 1.5 mL of methanol was added into 40 µL of the plasma sample, followed by the addition of 3 mL of CHCl_3_ and incubation for 1 h at room temperature with occasional vortex mixing. Then, 1.25 mL of purified water was added and allowed to stand for 10 min to facilitate the phase separation. The sample was centrifuged at 1000× *g* for 10 min at 4 °C, and aliquots of 2000 µL of the organic phase were collected. Finally, the aliquots were vacuum-dried and stored at −80 °C until further analysis.

#### 2.4.2. UPLC-QTOF/MS

The lipid extracts were reconstituted with 250 μL of an isopropanol-acetonitrile-water (2:1:1), and SYNAPT G2 QTof (Waters MS Technologies, Manchester, UK) was used for the UPLC-QTOF/MS analysis. The parameters of the mass spectrometer for positive ionization mode detection were as follows: Desolvation gas flow of 900 L/h, desolvation temperature of 550 °C, cone gas flow of 15 L/h, source temperature of 120 °C, capillary voltage of 2.8 kV, cone voltage of 40 V, and TOF-MS scan range of 100–2000 m/z. The data acquisition rate was set to 1.2 s with a 0.02-s interscan delay using the Waters MSE acquisition mode, and the full exact masses were collected simultaneously by rapidly alternating between two functions. Function 1 acquired data with a low collision energy of 4 and 2 eV for trap and transfer collision cells, whereas function 2 acquired data using a transfer collision energy ramp from 15 to 35 eV. All analyses were performed using a LockSpray to ensure the accuracy and reproducibility. Leucine–enkephalin was used as the lock mass at a concentration of 1 ng/μL and a flow rate of 5 μL/min. Data were collected in a continuum mode, and the LockSpray frequency was set at 20 s. All the data acquisition was controlled by the Waters MassLynx v4.1 software. The data was obtained from six individual samples, and each data is triplicated.

#### 2.4.3. Lipid Identification

The raw data were imported into the Progenesis QI software (Waters Corporation, MA, USA) for alignment. Further, the peak picking and identification of polar lipids were carried out using a high-resolution positive-ion MS, and the absolute intensities of all identified compounds were recalculated to determine the relative abundances and to normalize the values of the lipid molecules. The data were then exported into the EZinfo 2.0 software (Sartorius Stedim Biotech, Umeå, Sweden) for the multivariate statistical analysis, and the principal component analysis (PCA) and orthogonal projections to latent structures discriminant analysis (OPLS-DA) were used to create the final statistical models to obtain the group clusters. Lipid molecules with the strongest effect on the group clustering were identified as those with a VIP greater than one. In addition, potential ions with *p* < 0.05 and FC > 2 were selected for further metabolite relationship pathway characterization using the MetaboAnalyst web server. Human Metabolome Database (HMDB) IDs were matched with the Kyoto Encyclopedia of Genes and Genomes (KEGG) IDs for the KEGG mapping. IDs without a match were excluded from the analysis, and the Rattus norvegicus (rat) pathway library in the KEGG was selected for analysis.

### 2.5. Statistical Analysis

Data are expressed as the mean ± standard error (SE). Statistical differences were analyzed through the one-way analysis of variance (ANOVA) with Duncan’s test using the SPSS version 20.0 (SPSS Inc., Chicago, IL, USA). Differences were considered statistically significant where *p* < 0.05.

## 3. Results

### 3.1. Effect of RB on Body Weight and Masses of Liver, Omental Fat, and Epididymal Fat in Rats with HED-Induced Obesity

The effects of RB on the body weight and the masses of the liver tissue, omental fat, and epididymal fat are presented in Table 1. After eight weeks, the body weight and the masses of liver, omental, epididymal, and total fat of the HED groups significantly increased compared with the vehicle group until end of the study (*p* < 0.05). Although the initial body weight did not significantly differ among the groups, the final body weight was lower in the 1.0X (466 ± 8 g) and 2.0X (463 ± 5 g) groups by 4% and 5%, respectively, compared with the HED group (487 ± 5 g) (*p* < 0.05). However, the daily energy intake did not significantly differ between the HED and RB-0.5X, RB-1.0X, or RB-2.0X groups. The fat around the omental and epididymis of the vehicle, HED, RB-0.5X, RB-1.0X, and RB-2.0X groups weighed 3.74 ± 0.33, 7.03 ± 0.57, 6.40 ± 0.40, 7.10 ± 1.14, and 6.48 ± 0.64 g, respectively. The fat around the epididymis of the vehicle, HED, RB-0.5X, RB-1.0X, and RB-2.0X groups weighed 3.33 ± 0.36, 7.56 ± 0.53, 6.62 ± 0.26, 6.75 ± 0.75, and 6.96 ± 0.65 g, respectively. The omental and epididymal fat masses were greater in the HED group than the vehicle group by 88% (*p* < 0.05) and 127% (*p* < 0.05), respectively. However, the RB groups exhibited slightly lower epididymal fat compared with the HED group.

### 3.2. Effect of RB on Serum Biochemical Parameters of Rats with HED-Induced Obesity

The effects of RB on the uric acid, GOT, FFA, TC, TG, HDL, LDL, and glucose are presented in Table 2. The HED-fed SD rats had significantly higher serum levels of the uric acid, FFA, TC, TG, and glucose by 113% (*p* < 0.05), 68% (*p* < 0.05), 27% (*p* < 0.05), 288% (*p* < 0.05), and 123% (*p* < 0.05), respectively, compared with the SD rats fed with a normal diet. Significantly lower serum levels of the uric acid (30%, 41%, and 59% lower in the RB-0.5X, RB-1.0X, and RB-2.0X groups, respectively) and glucose (27%, 17%, and 17% lower in the RB-0.5X, RB-1.0X, and RB-2.0X groups, respectively) were exhibited by the RB-treated groups compared with the HED group.

### 3.3. Effect of RB on Hepatic Lipid Accumulation in Rats with HED-Induced Obesity

Liver tissue biopsies performed through the hematoxylin and eosin staining revealed that rats fed with an HED for eight weeks developed a higher degree of steatosis and microvesicular fatty changes. The extent of steatosis was diminished by the RB intervention (Figure 1A). In addition, the HED-fed to rats accumulated more hepatic TG (49% higher compared with the vehicle group, *p* < 0.05) and hepatic TC (102% higher compared with the vehicle group, *p* < 0.05). In the liver tissues of the RB-0.5X, RB-1.0X, and RB-2.0X groups, the TG (10.0 ± 1.4, 7.5 ± 1.1, and 7.2 ± 1.0 mg/g, respectively) and TC (1.0 ± 0.1, 1.0 ± 0.1, and 0.8 ± 0.0 mg/g) levels were significantly lower than those (12.7 ± 2.4 and 1.7 ± 0.1 mg/g, respectively) of the HED only group (*p* < 0.05) (Figure 1B and 1C). Thus, rats with HED-induced obesity fed RB-0.5X, RB-1.0X, and RB-2.0X diets accumulated significantly less hepatic TG and TC (21%, 41%, 43%, and 41%, 41%, 53%, respectively) compared with those fed only HED. Thus, strong dose-dependent effects were observed of the hepatic TG and TC levels. These results suggest that RB may have an inhibitory effect on lipid accumulation in the liver.

### 3.4. Effect of RB on Adipocyte Size Distribution in Rats with HED-Induced Obesity

Rats in the intervention groups were significantly heavier than those in the vehicle group. However, the body weights of the rats in the RB treatment groups increased more slowly than those of rats in the HED group (Figure 2A). Histological analyses revealed larger white adipocytes around the epididymis of the HED group compared with the vehicle group (Figure 2B). Nevertheless, the epididymal white adipocytes of the rats in the RB treatment groups were smaller than those in the HED group (Figure 2B). The results indicate that the RB-0.5X, RB-1.0X, and RB-2.0X intervention reduced the HED-induced adiposity.

### 3.5. Effect of RB on Lipidomics in Rats with HED-Induced Obesity Analyses of Lipidomic Profiles

PCA and supervised OPLS-DA score plots were used to determine which dominant lipid species were reduced by the RB amelioration of HED-induced obesity in rats. The vehicle and HED groups were compared to determine the obesity-related changes in serum lipids. A distinct clustering of the vehicle and HED groups was observed in the plots (Figure 3), suggesting that an HED induced severe lipid biosynthesis in SD rats. Therefore, we selected 29 variables determined based on the three thresholds (*p* < 0.05, VIP > 1, and FC > 2), among which eight lipid species were significantly increased by an HED: phosphotidylcholine (PC; 18:2/22:6), diacylglycerol (DG; 18:2/16:0), DG (18:2/18:1), DG (18:1/16:0), cholesteryl ester (CE; 20:5), CE (28:2), TG (18:0/16:0/18:3), and glycerol-1-2-hexadecanoate 3-octadecanoate. The PCA and OPLS-DA score plots were used to determine changes in the serum lipids due to the RB effect, revealing that the HED and RB-2.0X groups (Figure 4) were well separated. The results indicate that the RB treatment led to a limited reduction of the increment in two lipid species induced by the HED, namely DG (18:2/18:1/0:0) and TG (18:0/16:0/18:3).

### 3.6. Metabolic Pathway Analysis

The identified metabolites were exported to the MetaboAnalyst web server (https://www.metaboanalyst.ca/) for the metabolic pathway analysis to determine the obesity-related metabolic pathways involved. The pathway analysis results indicated that these metabolites were involved in the linoleic acid metabolism, alpha-linolenic acid metabolism, glycerophospholipid metabolism, steroid biosynthesis, and arachidonic acid metabolism.

## 4. Discussion

The abundance of phytonutrients in the RB has made RB a potential functional food for disease prevention [9]. Our results revealed that dietary RB supplementation provided a beneficial effect of the ameliorating body weight, morphometry, and biochemical alternations related to diet-induced obesity. The serum lipidomic changes revealed potential biomarkers, namely PC (18:2/22:6), DG (18:2/16:0), DG (18:2/18:1), DG (18:1/16:0), CE (20:5), CE (28:2), TG (18:0/16:0/18:3), and glycerol-1-2-hexadecanoate 3-octadecanoate for understanding HED-induced obesity. The RB treatment presented an anti-obesity effect by partially reducing the HED-induced DG (18:2/18:1) and TG (18:0/16:0/18:3) increases in obese rats.

Obesity is a major risk factor for the development of chronic diseases, such as type 2 diabetes and cardiovascular diseases. Excess consumption of high-energy-density food, such as eating a high-fat or high-sugar diet, leads to an increase in the white adipose tissue, which may cause metabolic, hormonal, and inflammatory changes resulting in organ damage [17]. Our study revealed that the HED intervention group had higher body weight as well as serum concentrations of uric acid, FFA, TC, TG, and glucose. Moreover, the morphometric analysis revealed liver steatosis and enlargement of adipocytes due to the HED.

RB is a rich source for dietary fiber, oligosaccharides, hemicelluloses, and nonstarchy polysaccharides as well as some water-soluble phytochemicals that may be beneficial to health [18,19]. A previous clinical study revealed that the RB supplementation could reduce blood glucose and lipid concentrations [20]. We found that the RB-1.0X and RB-2.0X treatment groups had lower HED-induced elevation of body weight and blood glucose and less hepatic accumulation of TC and TG, indicating that the RB supplementation had a protective effect against the HED-induced obesity. However, the increase in the rice bran dose (from RB-1.0X to RB-2.0X) has little effect and that the efficacy of rice bran is therefore not dose dependent. The main bioactive compound in RB is γ-oryzanol, which has demonstrated antioxidative, anti-inflammatory, antidiabetic, and anticancer effects [21,22]. Moreover, γ-oryzanol tends to reduce high-fat-diet-induced cholesterol accumulation in the liver, and our findings supported this [21]. Thus, we propose that RB tends to reduce the accumulation of lipids in the liver because of its major bioactive compound, γ-oryzanol.

We investigated the serum lipidomic profiles of SD rats with the HED-induced obesity using HPLC-QTOF-MS. To avoid bias, an untargeted analysis was carried out to obtain novel biological insight regarding the lipid species affected by HEDs [23]. High blood lipids were found to be associated with obesity [24]. We found that concentrations of CE, PC, DG, and TG were increased in the HED group compared with the vehicle group. The PC acyl chain length was found to be associated with the TC and hepatic cholesterol concentrations [23], whereas the long chain of the PC was found to be positively associated with the glucose, insulin, and leptin concentrations [25]. CE was associated with liver cholesterol homeostasis [26]. Our data revealed that SD rats with HED-induced obesity exhibited significant increases in the lipid accumulation and glucose levels in the liver. Thus, we propose that PC and CE are both involved in the glucose and cholesterol regulation. Obesity has been associated with glucose intolerance and insulin resistance in animal [27,28] and human studies [29,30,31,32], suggesting that lipids play an important role in modulating energy metabolism in the body. An increase in the concentration of DG may decrease liver insulin sensitivity and therefore lead to fat accumulation in hepatocytes [33]. Similarly, in our study, the obese rats of the HED group had a higher concentration of DG than their lean littermates. Lipid droplet infiltration in hepatocytes detected through the liver resection and the elevation in hepatic TG and TG revealed consistent results. However, the RB intervention groups exhibited lower concentrations of blood glucose and accumulation of hepatic lipids than the HED group, and a significantly lower serum DG in the RB-2.0X group was observed. Thus, these results reveal that the RB supplementation can ameliorate adiposity by partially reducing HED-induced DG and TG increases in obese rats.

## 5. Conclusions

In conclusion, rice bran could act as an anti-obesity supplement in rats, as demonstrated by partially reducing the HED-induced DG and TG increases in obese rats, and thus limit the metabolic diseases associated with obesity and accumulation of body fat and hepatic lipids in rats.

## Figures and Tables

**Figure 1 nutrients-11-02033-f001:**
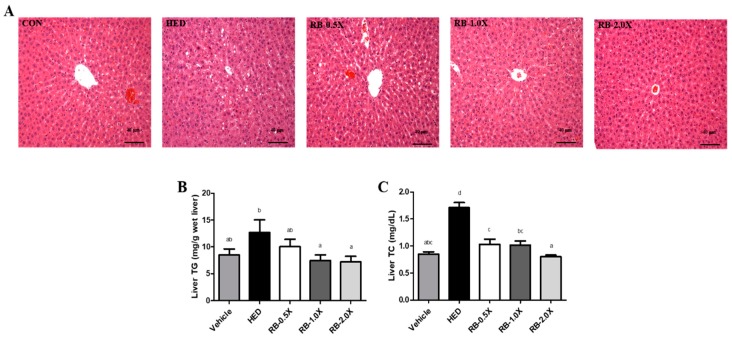
Effect of rice bran (RB) on the hepatic lipid accumulation in rats with high-energy diet (HED)-induced obesity. (**A**) Hematoxylin and eosin staining of liver sections (100× magnification). Scale bar: 40 μm. (**B**) Liver triglyceride (TG) concentration and (**C**) liver total cholesterol (TC) concentration. The x-axis parameters are as follows: Vehicle, SD rat fed with chow diet; HED, SD rat fed with HED; RB-0.5X, SD rat fed with HED and RB-0.5X; RB-1.0X, SD rat fed with HED and RB-1.0X; and RB-2.0X, SD rat fed with HED and RB-2.0X. The data are expressed as the mean ± SE (*n* = 10). Different letters (a, b, and c) indicated significant differences at *p* < 0.05 by one-way ANOVA.

**Figure 2 nutrients-11-02033-f002:**
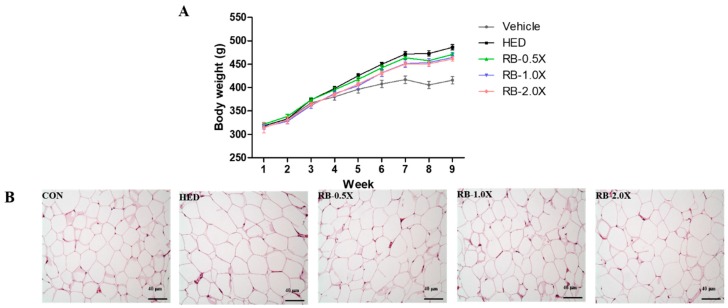
Effect of RB on body weight and epididymal white adipose tissue (eWAT) change in rats with HED-induced obesity. (**A**) Body weight change over time; (**B**) hematoxylin and eosin staining of eWAT sections (×100 magnification). Scale bar: 40 μm. Vehicle, SD rat fed chow diet; HED, SD rat fed HED; RB-0.5X, SD rat fed HED with RB-0.5X; RB-1.0X, SD rat fed HED with RB-1.0X; RB-2.0X, SD rat fed HED with RB-2.0X. The data are expressed as the mean ± SE (*n* = 10).

**Figure 3 nutrients-11-02033-f003:**
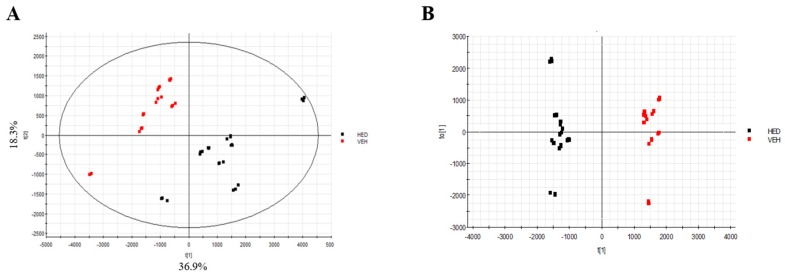
Score plots of the serum samples from the vehicle and HED groups. (**A**) principal component analysis (PCA) score plot and (**B**) orthogonal projections to latent structures discriminant analysis (OPLS-DA) score plot. Vehicle, SD rat fed chow diet; HED, SD rat fed HED.

**Figure 4 nutrients-11-02033-f004:**
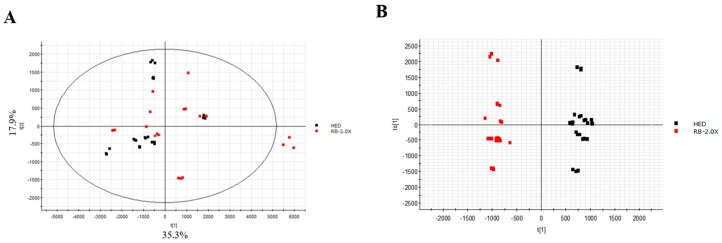
Score plots of the serum samples from the HED and RB-2.0X groups. (**A**) PCA score plot and (**B**) OPLS-DA score plot. HED, SD rat fed HED; RB-2.0X, SD rat fed HED with RB-2.0X.

**Table 1 nutrients-11-02033-t001:** Comparison of the physical and metabolic effects of rice bran (RB) in rat groups.

	Vehicle	HED	RB-0.5X	RB-1.0X	RB-2.0X
Water (mL)	45.3 ± 0.8 ^b^	26.3 ± 0.4 ^a^	25.8 ± 0.2 ^a^	26.7± 0.3 ^a^	25.0 ± 0.3 ^a^
Food intake (g/day)	23.3 ± 0.4 ^a^	23.5 ± 0.4 ^a^	23.6 ± 0.4 ^a^	23.0 ± 0.3 ^a^	23.2 ± 0.4 ^a^
Energy intake (kcal/day)	77.9 ± 1.3 ^a^	99.2 ± 1.6 ^b^	99.9 ± 1.9 ^b^	98.2 ± 1.5 ^b^	99.4 ± 1.7 ^b^
Feed efficiency ^1^	0.5 ± 0.2 ^a^	0.9 ± 0.2 ^a^	0.8 ± 0.2 ^a^	0.8 ± 0.2 ^a^	0.8 ± 0.1 ^a^
Final body weight (g)	416 ± 8 ^a^	487 ± 5 ^c^	472 ± 4b ^c^	466 ± 8 ^b^	463 ± 5 ^b^
Body weight gain (g)	97.1 ± 4.4^a^	168.1 ± 5.4 ^b^	149.1 ± 2.5 ^b^	148.4 ± 6.1 ^b^	146.8 ± 14.3 ^b^
Liver	11.55 ± 0.41 ^a^	15.34 ± 0.28 ^b^	14.09 ± 0.23 ^b^	14.09 ± 0.23 ^b^	14.24 ± 0.42 ^b^
Omental fat	3.74 ± 0.33 ^a^	7.03 ± 0.57 ^b^	6.40 ± 0.40 ^b^	7.10 ± 1.14 ^b^	6.48 ± 0.64 ^b^
Epididymal fat	3.33 ± 0.36 ^a^	7.56 ± 0.53 ^b^	6.62 ± 0.26 ^b^	6.75 ± 0.75 ^b^	6.96 ± 0.65 ^b^
Total fat	9.51 ± 0.88 ^a^	24.40 ± 1.50 ^b^	22.65 ± 0.76 ^b^	23.76 ± 1.81 ^b^	23.36 ± 1.80 ^b^

Data are expressed as the mean ± standard error (SE) (*n* = 10). Different letters (a, b, and c) indicate significant differences at *p* < 0.05 in the one-way ANOVA. Column titles indicate the following: Vehicle, Sprague–Dawley (SD) rat fed chow diet; high-energy diet (HED), SD rat fed HED; RB-0.5X, SD rat fed HED with RB-0.5X; RB-1.0X, SD rat fed HED with RB-1.0X; RB-2.0X, SD rat fed HED with RB-2.0X. ^1^ Feed efficiency = weight gain/food intake.

**Table 2 nutrients-11-02033-t002:** Effect of RB on the serum biochemical parameters in rats with high-energy diet (HED)-induced obesity.

	Vehicle	HED	RB-0.5X	RB-1.0X	RB-2.0X
Uric acid (mg/dL)	3.0 ± 0.4 ^a^	6.4 ± 0.6 ^c^	4.5 ± 0.2 ^b^	3.8 ± 0.6 ^ab^	2.6 ± 0.3 ^a^
GOT (U/L)	64 ± 2 ^a^	66 ± 2 ^a^	63 ± 2 ^a^	62 ± 1 ^a^	64 ± 1 ^a^
FFA (mmol/L)	0.62 ± 0.03 ^a^	1.04 ± 0.06 ^b^	0.95 ± 0.07 ^b^	0.88 ± 0.09 ^b^	1.03 ± 0.07 ^b^
TC (mg/dL)	59 ± 4 ^a^	75 ± 4 ^b^	72 ± 5 ^b^	68 ± 4 ^ab^	69 ± 6 ^ab^
TG (mg/dL)	40 ± 3 ^a^	155 ± 13 ^b^	135 ± 13 ^b^	137 ± 16 ^b^	133 ± 10 ^b^
HDL (mg/dL)	35.3 ± 2.3 ^a^	35.4 ± 0.8 ^a^	37.8 ± 2.5 ^a^	34.8 ± 1.8 ^a^	36.1 ± 2.6 ^a^
LDL (mg/dL)	14.0 ± 1.1 ^a^	11.5 ± 0.8 ^a^	11.8 ± 1.1 ^a^	11.4 ± 1.1 ^a^	11.4 ± 1.1 ^a^
Glucose (mg/dL)	153 ± 8 ^a^	341 ± 25 ^c^	250 ± 14 ^b^	283 ± 24 ^b^	283 ± 8 ^b^

Data are expressed as the mean ± SE (*n* = 10). Different letters (a, b, and c) indicate significant differences at *p* < 0.05 in one-way ANOVA. Vehicle, SD rat fed chow diet; HED, SD rat fed HED; RB-0.5X, SD rat fed HED with RB-0.5X; RB-1.0X, SD rat fed HED with RB-1.0X; RB-2.0X, SD rat fed HED with RB-2.0X.

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
