# Peer review of "Rice Bran Reduces Weight Gain and Modulates Lipid Metabolism in Rats with High-Energy-Diet-Induced Obesity"

_nutrients, 2019, doi:10.3390/nu11092033_

Round 1

Reviewer 1 Report

The paper "Rice Bran Prevents Obesity and Modulates Lipid Metabolism in Rats with High-Energy-Diet-Induced Obesity" Yang et al., discusses the potential use of Rice Bran (a by-product of rice milling) to reduce obesity in Rats.

The paper is topical due to the obesity epidemic.

However there are a few issues which I feel need to be adressed before consideration for publication:

There is no mention of the LC method used for the analysis. Did the authors normalize the MS signals? There is a descripancy in the number of samples. The PCA/OPLS-DA plot show more than 10 points for analysis in each group when n=10. There is no clear seperation in PCA for the HED and RB-2.0X. Did the R2 and Q2 values indicate a good model for OPLS-DA? Were there differences in the other 27 lipid species by RB treatment? Also a plot showing the percent decrease of the two lipid species along with statistical significance would be helpful. Minor issue: There are formatting issues in the print (Lines 208 to 219)

Author Response

Reviewer #1

Q1. There is no mention of the LC method used for the analysis.

Answer: Thanks for reviewer suggestion. In this study, the method used to analyze serum lipidomic change is UPLC-QTOF/MS and the method is mentioned in the revised manuscript. (Page 3 Line 123-137)  

Q2. Did the authors normalize the MS signals?

Answer: Thanks for reviewer suggestion. We did the normalization of the MS signals. The detail has been mentioned in our revised manuscript. (Page 4, Lines 141-142)

Q3. There is a descripancy in the number of samples. The PCA/OPLS-DA plot show more than 10 points for analysis in each group when n=10.

Answer: Thanks for reviewer suggestion. In this study, the data was obtained from 6 individuals sample, and each data is triplicated. It has been described clearly in our revised manuscript. (Page 3, Lines 136-137)

Q4. Did the R2 and Q2 values indicate a good model for OPLS-DA?

Answer: The range of R2 is in between 0 and 1, the higher level, the higher predictive accuracy. In the present study, our R2 is 0.98 while Q2 is 0.97. According to Chin (1998) and Henseler et al. (2009), R2 value greater than 0.67 indicate a high predictive accuracy, a range of 0.33 - 0.67 indicated a moderated effect, R2 between 0.19 and 0.33 indicate low effect, while the R2 value below 0.19 considered unacceptable (the exogenous variables unable to explain the endogenous dependent variable). While Q2 value of greater than zero for a particular reflective endogenous latent variable indicate the path model’s predictive relevance for a specific dependent construct (Hair et al. 2016). We believed that the analysis model has a high accuracy.

Reference

WW Chin. Issues and Opinion on Structural Equation Modeling. Commentary. 1998, 22, vii-xvi.

Q5. Also a plot showing the percent decrease of the two lipid species along with statistical significance would be helpful.

Answer: Thanks for reviewer suggestion. In the present study, we only focused in the 2 lipid species, DG (18:2/18:1/0:0) and TG (18:0/16:0/18:3), which significantly reversed the increment induced by HED. The detail has been modified in our revised manuscript. (Page 7, Line 225-227)

Fatty acid

VEH

HED

RB-2.0X

DG

(18:2/18:1/0:0)

27151 ± 1684a

128775 ± 4902b

3631 ± 280.9c

TG

(18:0/16:0/18:3)

103628 ± 5461a

274313 ± 15800b

129 ± 5.7c

Q6. There are formatting issues in the print (Lines 208 to 219).

Answer: It has been modified in our revised manuscript. (Page 7, Lines 216-227)

Reviewer 2 Report

Manuscript by Ching Yang demonstrated that the Rice Bran Prevents Obesity induced by High-Energy-Diet by Modulating the Lipid Metabolism in Rats.

In the literature, studies on humans and other in vivo experiments, people have demonstrated the health beneficial properties of rice bran oil. It has shown to be improving the risk factors associated with cardiovascular disease (PMID: 23798921) by decreasing the levels of TC, LDL, and atherogenic ratio of TC / HDL. Rice bran oil terpenoids (ferulic acid esters of phytosterols and triterpenoids) improves postprandial hyperglycemia in healthy humans (PMID: 30302087).  Ferulic Acid and γ-Oryzanol inhibited hepatic fat accumulation and inflammation induced by High-Fat and High-Fructose Diet in rats (PMID: 25646799). Type 2 diabetes, obesity and cardiovascular diseases are interconnected and causes by metabolic dysfunctions that includes hyperglycemia, hypercholesterolemia, hypertriglyceridemia and insulin resistance (IR).

Line 28: by product

Line 57: More recent citations are required in the introduction section. For eg: anti-obesity drugs withdrawal, see and include updated and recent articles, PMID: 27894343 and PMID: 25114779. Health beneficial properties of RBO also required more citations appropriately which supports the current study.

Line 99: Have authors extracted FFA before its measurement by auto analyzer? Content should be in detail. Authors can provide the extraction and measurement methodologies for all the parameters as supplementary material.

Line 112: Lipid extraction section; Ideal temperature of sample incubation would be at 4C, rather than room temperature. Lipids are highly sensitive to light and temperature.

Line 113: Either reference 14, has not described the determination of TC and TG in detail. In the reference authors have mentioned that they were measured TC by C1600 analyzer and TG by Beckman DX 800. Did authors used Beckman DX 800 for TG measurement? Detail description would be appropriate or correct citations are required.

Lines 238-250: Did authors observed only particular lipids in each class of lipids such as DG (18:2/18:1), DG (18:1/16:0) in class DG? Not whole class? Like the average of all DGs or TGs?  

Author Response

Reviewer #2

Q1. Line 28: by product.

Answer: It has been modified in our revised manuscript. (Page 1, Line 28)

Q2. More recent citations are required in the introduction section. For eg: anti-obesity drugs withdrawal, see and include updated and recent articles, PMID: 27894343 and PMID: 25114779. Health beneficial properties of RBO also required more citations appropriately which supports the current study.

Answer: Thanks for reviewer suggestion. The references have been added in our revised manuscript. (Page 2, Lines 55-57; References 5 and 6)

Reference

Cheung BC.; Cheung TT.; Samaranayake NR. Safety of antiobesity drugs. Europe PMC. 2013, 4, 171–181.

Onakpoya IJ.; Heneghan CJ.; Aronson JK. Post-marketing withdrawal of anti-obesity medicinal products because of adverse drug reactions: a systematic review. 2016, 14, 191.

Q3. Have authors extracted FFA before its measurement by auto analyzer? Content should be in detail. Authors can provide the extraction and measurement methodologies for all the parameters as supplementary material.

Answer: We appreciate the reviewer’s comments regarding to the FFA measurement. In current study, the Clinical laboratory with ISO15189-TAF certification was commissioned for all the serum parameters analysis by auto analyzer. We had inquired the FFA measurement protocol for auto analyzer application and it basically depend on the enzymatic method for oxidative products development, such like MEHA, and reaction.

Reference

https://www.quimigen.com/-186/non-esterified-fatty-acid-assay-415001190.html

Principle: NEFA in the sample becomes to fatty acyl CoA, AMP and pyrrole phosphoric acid due to the effect of Fatty-acyl-CoA Synthase in presence of CoA and ATP. The generated fatty acyl CoA oxidized by acyl-CoA oxidase to generate 2, 3-trans-enoyl-CoA and H2O2. Then MEHA formed from H2O2 under peroxide action. The oxidative polymerization occurs between the MEHA and 4-amino-antipyrine to produce a colored pigment. At a specific wavelength, NEFA concentration in the sample can be calculated by comparing the calibrator result with same processing.

We believed the samples could be precisely measured if the model or sample preparation is appropriate. Besides, the professional facility (clinical laboratory and Biotechnology Company) could also assist the experimental accuracy and efficiency. The analysis methods should be depended on the different experimental purpose and sample types.

Q4. Lipid extraction section; Ideal temperature of sample incubation would be at 4C, rather than room temperature. Lipids are highly sensitive to light and temperature.

Answer: Thanks for reviewer suggestion. Based on the Folch method, room temperature is the ideal temperature for lipid extraction with higher recoveries rate. The reference has been added in our revised manuscript. (Reference 15)

Reference

Folch J.; Lees M.; Sloane Stanley G.H. A simple method for the isolation and purification of total lipides from animal tissues. J. Biol. Chem. 1957, 226, 497-509

Q5. Either reference 14, has not described the determination of TC and TG in detail. In the reference authors have mentioned that they were measured TC by C1600 analyzer and TG by Beckman DX 800. Did authors used Beckman DX 800 for TG measurement? Detail description would be appropriate or correct citations are required.

Answer: We appreciate the reviewer’s comments regarding the details of TG and TC measurement. We will incorporate the detail descriptions for liver TC and TG measurements in our revised manuscript. (Page 3, Lines 101-108)

Q6. Did authors observed only particular lipids in each class of lipids such as DG (18:2/18:1), DG (18:1/16:0) in class DG? Not whole class? Like the average of all DGs or TGs? 

Answer: Thanks for reviewer suggestion. In this study, we observed whole class of lipid species but only focused in the most significantly different lipid species, which are DG (18:2/18:1/0:0) and TG (18:0/16:0/18:3). The detail has been modified in our revised manuscript. (Page 7, Lines 216-227)
